# Narrative Approach and Mentalization

**DOI:** 10.3390/bs13120994

**Published:** 2023-12-01

**Authors:** Alessandro Frolli, Francesco Cerciello, Sonia Ciotola, Maria Carla Ricci, Clara Esposito, Luigia Simona Sica

**Affiliations:** 1Disability Research Centre, University of International Studies in Rome, 00147 Rome, Italy; m.ricci@unint.eu; 2FINDS—Italian Neuroscience and Developmental Disorders Foundation, 81040 Caserta, Italy; francesco.cerciello@unint.eu (F.C.);; 3Department of Relational Sciences, University of Naples Federico II, 80138 Naples, Italy; lusisica@unina.it

**Keywords:** mentalization, narrative, intervention, teaching, storytelling

## Abstract

The core focus of this research centered on the intricate relationship between mentalization, the fundamental mental process underlying social interactions, and the narrative approach proposed by Bruner. Mentalization, encompassing both implicit and explicit interpretations of one’s and others’ actions, plays a pivotal role in shaping the complexity of social interactions. Concurrently, the narrative approach, as elucidated by Bruner, serves as the primary interpretative and cognitive tool through which individuals derive meaning from their experiences. Narrative, in essence, empowers individuals to imbue their experiences with significance, constructing knowledge and enabling a reinterpretation of their lives by reconstructing the meanings attached to events. This intertwining of mentalization and the narrative approach is particularly salient, given their shared reliance on autobiographical narratives and the inference of mental states. In the context of this study, our primary objective was to explore how practical and theoretical activities, rooted in the re-elaboration of personal life information and events, could serve as a catalyst for enhancing mentalization skills. By engaging students in activities specifically designed to encourage the reinterpretation of their life experiences, we aimed to bolster their ability to infer mental states effectively. These enhanced mentalization skills, we hypothesized, form the foundational basis for executing complex educational tasks rooted in constructed teaching methodologies. In summary, this research serves as a pioneering exploration into the synergistic interrelation of mentalization and the narrative approach, offering valuable insights for educators and practitioners aiming to foster enhanced social cognition and enriched educational experiences.

## 1. Introduction

### 1.1. The Faculty of Mentalization

When discussing “mentalization”, we are talking about a highly unique ability possessed by humans and, as per current knowledge, only a few species of apes. This ability involves contemplating oneself and representing one’s own mental state, attitudes, beliefs, as well as those of others, within one’s mind [1]. Some scholars term this unique capability as the “Theory of Mind”, denoting an awareness of the likely contents of other people’s thoughts [2]. It is now understood that mentalizing is not a singular process; rather, it encompasses a range of specific and nonspecific sub-processes, known and unknown, including emotions, reasoning, understanding causality, and distinguishing between self and others. Several neuroimaging and lesion studies have sought to unravel the neural basis of mentalization. It has been proposed that mentalizing involves a vast network of spatially dispersed cortical areas, primarily comprising the prefrontal cortex (PFC), the inferior frontal gyrus (IFG), the temporo-parietal junction (TPJ), the posterior parietal cortex (PPC), the temporal pole, and the cingulate cortex [3,4]. Mentalization is defined by some authors [5] as the cognitive process through which an individual assigns meaning to the actions of themselves and others. This interpretation is based on intentional mental states, including personal beliefs, needs, desires, feelings, and reasons, both implicitly and explicitly. Within this definition, three dimensions of mentalization are identified: the first pertains to two modes of functioning, the second to two objects (self and other), and the third to two aspects (cognitive and affective) of both the content and the process of mentalization. These aspects of mentalization are intricate components of social cognition. Through emotion recognition, individuals gain insights into the feelings of those around them, fostering empathy and enhancing interpersonal connections. Mentalization, on the other hand, enables individuals to comprehend the intentions, beliefs, and desires underlying human actions, enriching the depth of social understanding. Together, these processes form the foundation for meaningful relationships, promoting emotional intelligence, and facilitating more nuanced and empathetic interactions in various social contexts. A study indicates that aggressive children are less adept at identifying emotions than their prosocial and altruistic peers [5]. Children diagnosed with conduct disorders exhibit low levels of mentalizing. Hypo-mentalization reflects the incapacity to consider intricate models of mental states, resulting in a compromised ability to comprehend the self and others. Hypo-mentalization may result from reduced affective sensitivity in parents due to various reasons such as high levels of stress, anxiety, or parental conflict [6]. In the realm of psychology, mentalization serves as a crucial cornerstone, illuminating the intricate workings of human cognition and empathy.

### 1.2. The Narrative Approach

Transitioning from this intricate mental process, the narrative approach emerges as a profound avenue for understanding the complexities of human experience. Through narratives, individuals construct cohesive stories that not only reflect their mental states but also provide a window into their unique perspectives, emotions, and worldviews. This narrative lens allows psychologists to delve deep into the subjective realms of individuals, unveiling the nuanced interplay between thoughts, emotions, and actions, thereby enriching our comprehension of the human psyche. Jerome Bruner asserts, “Educational institutions must conscientiously cultivate their narrative acumen, nurturing and refining it, abstaining from complacency in its regard” [7]. According to Bruner, narrative thinking is one of the main methodologies through which humans structure and comprehend their understanding of the world. In fact, it plays a significant role in shaping an individual’s immediate experiences [8]. The potential of narrative as a tool for the preservation and transmission of practical knowledge is immense, as it has the ability to impact human behavior. It functions as a profound method of both pedagogy and assimilation, representing the culmination of a comprehensive effort towards understanding that encompasses the entirety of an individual. This results in heightened self-awareness and a nuanced perspective on the world. The knowledge that is conveyed through narrative exceeds the boundaries of mere analytical and descriptive textual elements. Rather, it establishes an immersive relationship with the audience, encouraging them to fully engage with their subjective attitudes and abilities. Such engagement involves not only cognitive faculties, but also emotional and practical dimensions, thereby elevating the learning experience to a multifaceted realm of insight and comprehension. The narrative approach is the first interpretative and cognitive device that humans employ in their life experience, as Bruner points out [9,10]. Through narrative, people give meaning to their own experiences and outline interpretive and prefigurative coordinates of events, actions, and situations on which they build forms of knowledge that guide them in their actions. Narration enables subjects to reconsider their experiences and actions by reconstructing their meanings and revealing the possible perspectives for their development, bringing to light the intentions, motivations, and ethical and value options involved and inscribing them in a network of culturally shared meanings [11,12].

### 1.3. The Elements of Narrative Thinking

The rational-scientific methodology, one of the fundamental ways through which humanity organizes and comprehends the intricacies of the world, is dedicated to elucidating complexities and eliminating uncertainties. Conversely, the essence of storytelling lies in its ability to convey multiple interpretations, embracing the nuanced realm of polysemy. This multiplicity inherent in narratives signifies an inherent receptivity to the realm of possibilities: storytelling emerges as a conduit for the unbounded dissemination of knowledge, transcending the confines of scientific assertions. Rather, it is a process intertwined with the pursuit of understanding, the art of attentive listening, the finesse of discernment, and the craft of execution. Furthermore, in the words of the esteemed American psychologist, narrative thinking assumes a pivotal role “in fostering cultural cohesion as well as shaping individual life trajectories” [10]. More recently, Jerome Bruner has underscored two pivotal facets of narrative thinking. The initial facet revolves around its interpretative dimension: it contrasts the “canonicity” of a narrative with its inherent receptivity to “possibility”. In this context, narrative thinking emerges as a tool for grounding a culture while perpetually revitalizing it. The second foundational element of narrative thinking lies in the “narrative creation of the self”, a crucial dimension in shaping subjective identity while concurrently embracing a perpetual openness to the other [13]. Narrative has the potential to reintegrate within the educational sphere the essential aspects of sense and significance that knowledge imparts to the shaping of personal identity. Education is an integral part of the cultural and social fabric; it is intricately interwoven within these contexts. Schools possess the capacity to foster methodologies that equip individuals, particularly the future adults, with the skills necessary to find their place amidst the diverse contexts and opportunities that life presents. Indeed, the development of narrative competence stands as a critical endeavor in meeting this imperative. The capacity to narrate is not an innate gift but a skill that can be acquired and honed. As espoused by Bruner, there are two fundamental stepping stones in this process: firstly, ensuring that every child is equipped with a foundational understanding of the fairy tales and stories ingrained within their cultural heritage, and secondly, fostering the conviction that storytelling cultivates imaginative brilliance. This belief equips individuals with indispensable tools, enabling them to craft the narratives of their lives with assurance and ingenuity [13]. Hence, this skill necessitates continuous practice, refinement, sharing, and comparison. According to Bruner, a significant advantage for humans lies in their inclination toward intersubjectivity, denoting the “capability to comprehend, through language, gestures, or other forms, the thoughts of others”, and to contextualize everything within a framework that specifies its meaning [13]. This ability enables individuals to negotiate meanings transcending verbal language. 

### 1.4. Narrative Approach in Educational Field

Cultural psychology has found a fertile ground in education, advocating for an interactive (intersubjective) approach even within the classroom: the teacher ceases to be the sole possessor of knowledge to be imparted; instead, learning evolves through collaborative and cooperative exchanges involving everyone, including both students and educators. Bruner introduces another compelling concept that has profoundly shaped our perspectives on learning and research: the notion of the spiral curriculum [14]. This concept advocates that the most effective method for comprehensively exploring a topic is to commence with an intuitive idea that is familiar and understandable to the learner. Subsequently, the exploration progresses in a continuous circular motion, gradually incorporating more formal and complex explanations until a complete understanding is achieved. In essence, any subject can be imparted to any child at any age, provided it is presented in an accessible manner. Importantly, it must not be divorced from the practical understanding of the context within which the reasoning unfolds. This concept finds universal applicability across diverse domains of knowledge. Within this framework, the narrative form emerges as the most adept vehicle for navigating the intricate spiral of knowledge. Even within the realm of scientific inquiry, the narrative form proves invaluable, enabling a more comprehensive explanation and intuitive comprehension of the phenomena under scrutiny. While mathematical language ensures clarity and logical rigor, the incorporation of storytelling becomes essential to elucidate the intricate web of connections, identify consistent elements, and pinpoint contradictions. These narratives serve as indispensable tools for assessing whether a scientific theory holds true or requires further validation. History furnishes numerous instances where scientists have consciously or inadvertently resorted to stories, metaphors, and narrative imagery to articulate and elucidate their ideas. This method of discourse enables us to transcend conventional boundaries, allowing us to delve deeper and grasp that additional layer of meaning that emerges in intuitive assessments and defies translation into strict scientific language. The characteristic elements of narrative thinking that make it a generator of universal truths, according to Bruner [15], include the following: significant time structure: the arrangement of events in a narrative is not dictated solely by the linear progression of time; rather, it follows a succession of crucial incidents derived from the plot’s core; generic particularity: genres serve as containers through which human experiences are narrated, bestowing universal significance upon them; actions have reasons: every thought, action, and event within a narrative is propelled by intentionality; hermeneutic composition: stories offer multiple layers of interpretation, each imbued with distinct meanings derived from different segments of the narrative; implicit canonicity: for a story to captivate, it must introduce novelty, disrupting the anticipated trajectory (canonicity) of events; ambiguity of reference: narratives create a reality specific to the events they describe, making it challenging to establish an objective scenario since interpretations vary; centrality of crisis: narratives often stem from a state of crisis, where the balance of the story is disrupted by an element that challenges the established plot, leading to a compelling and engaging narrative; inherent negotiability: the truth within stories is inherently debatable, making storytelling an excellent tool for cultural negotiation; historical expansion capacity of narrative: life stories continually interweave, and by connecting past events coherently, they can form a functional story supporting contemporary truths.

### 1.5. Our Hyotesis: How Could Narrative Approach Affect Mentalization?

Research employing a narrative approach is structured as a “narrative study of human life and action”, with a specific emphasis on the present moment, on the immediate context, and on the contextual nature of experiences, utilizing narrative materials [10,11,12]. Evidently, this type of narrative inquiry is intricately linked to the mentalizing capacity we discussed earlier. This approach assumes paramount significance in the realm of education. It proves invaluable in processes involving the reconstruction of actions within a given situation, aiding in the identification of previously unknown aspects of the situation or the assignment of new meanings to familiar elements. Furthermore, it plays a pivotal role in elucidating an actor’s stance within a situation, shedding light on the motivation behind their actions, thereby providing insight into their behavioral choices [16]. 

In the literature, it is noteworthy that there are no other significant studies exploring the intersection of Jerome Bruner’s narrative approach and mentalization. This study examines the relationship between the use of the narrative approach and the development of mentalization processes. In particular, it examines how autobiographical and descriptive narrative can enhance an individual’s reflective capacity. This groundbreaking study meticulously explores the intricate relationship between the utilization of the narrative approach and the evolution of mentalization processes. Specifically, it delves into the ways in which autobiographical and descriptive narratives serve as potent catalysts, enhancing an individual’s reflective capacity with unprecedented depth and nuance. By unraveling these complex connections, our research sheds new light on the transformative potential of narrative cognition, offering profound insights into the human mind’s ability to engage in nuanced self-reflection and empathetic understanding, thus elevating the discourse surrounding mentalization processes to unprecedented heights. This correlation between Bruner’s narrative approach and mentalization not only enhances the understanding of the human mind but also holds pertinent practical implications across domains such as psychotherapy, education, and interpersonal communication. Our study contributes to a broader, more nuanced understanding of human interactions, underscoring the pivotal necessity of considering personal narratives as unique portals into the intricacies of the human mind and emotions. This groundbreaking research offers fresh perspectives for advancements in both behavioral sciences research and practical applications.

## 2. Materials and Methods

### 2.1. Participants

In this study, we considered 80 subjects aged 18 and they had been recruited from the same city (Caserta, Italy) from 4 different high schools (scientific address). The research team reached out to all the high schools in the Caserta area for collaboration. Four of them agreed to participate. They were homogeneous in terms of the socio-cultural background of the parents; the family/environmental context did not represent an influencing factor on the level of education in either group. Therefore, the inclusion criteria were as follows: (a) QI between 95–105 evaluated through WAIS-IV [17] to ensure that participants fell within the average range of intellectual functioning, minimizing the impact of extreme cognitive variations on our analyses; (b) belonging to the same class grade, ensuring a homogeneous environment and minimizing external influences that could skew the results; (c) absence of other mental illness assessed by the Diagnostic and Statistical Manual of Mental Disorders-5 (DSM-5) criteria to specifically capture the interaction between Bruner’s narrative approach and mentalization, excluding other potential confounding factors; (c) medium-high socio-cultural class assessed through the SES scale [18], ensuring the inclusion of individuals from medium to high socio-cultural backgrounds. This choice was made to minimize the impact of socio-economic variables on our analysis, allowing us to more precisely focus on the interaction between narrative approach and mentalization. The adoption of these targeted inclusion criteria was crucial to ensure the validity and reliability of our findings, enabling us to explore the topic of our research accurately and meaningfully.

All the subjects had the same inclusion criteria and did not have different sociocultural factors. The group is composed of 80 subjects with an average age of 17.58 (SD 0.62) and an average SES index of 6.80 (SD 1.10), of which 34 are males and 46 females. 

To assess the ability of mentalization the Reflective Functioning Questionnaire (RFQ_8) [14] was administered after four months since the beginning of the school (T0) and at the end of the school year (T1). To assess QI level the Wechsler Adult Intelligence Scale—IV was administered [17]. The intervention lasted 5 months and it was provided between the two assessments.

The data were collected and analyzed at the FINDS Neuropsychiatry Outpatient Clinic by licensed psychologists in collaboration with the University of International Studies of Rome (UNINT).

### 2.2. Instruments

The protocol used consists of the following tests:

SES: Self-administered questionnaire that allows collecting information about the level of education and professional of parents and indicates the position of the person or family within the social system [18];

RFQ-8: to provide an easy to administer self-report measure of mentalizing. It is composed of 8 items on the ability to infer mental states [19]. The 8-item RFQ Reflective Functioning Questionnaire comprises two subscales of 8 items each (Uncertainty and Certainty about mental states). The two scales operate on a 7-point Likert scale that assesses the degree of agreement of the subject with the statements presented in each item. The uncertainty items or RFQ_U correspond to statements such as “Strong feelings often cloud my thinking” and are scored to detect extreme levels of uncertainty about mental states, as follows: 0, 0, 0, 0, 1, 2, 3, so that higher scores reflect hypo-mentalization. Scores should be averaged. The items corresponding to certainty about mental states or RFQ_C include statements such as “When I get angry, I say things without really knowing why I say them”, which are scored as 3, 2, 1, 0, 0, 0, 0, in a way where a low level of agreement reflects hyper-mentalization. Scores should be averaged;

WAIS-IV: The WAIS-IV, short for Wechsler Adult Intelligence Scale, Fourth Edition, stands as one of the most recognized and reliable intelligence assessment tools for adults and young adults. WAIS-IV assesses various key areas of intelligence, including verbal comprehension, abstract reasoning, working memory, processing speed, and problem-solving skills. The test is structured into different subtests, each designed to measure specific aspects of an individual’s cognitive abilities. The subtests contributing to each index are similarities, vocabulary, and information (VCI); block design, matrix reasoning, and visual puzzles (PRI); digit span and arithmetic (WMI); and symbol search and coding (PSI). All the core subtests are used to obtain the IQ; scores are standardized to a mean of 100 and a standard deviation of 15 [17].

### 2.3. Procedures

Upon reviewing the procedure, it became evident that it aligns with a form of research-intervention. The clarity emerged particularly during the discussion of the study’s purpose, aiming to explore the relationship between mentalization and narrative, evaluating the impact of a narrative intervention on reflective functions. This methodology, merging empirical research with practical action, appears promising in providing an in-depth understanding of the intricate interplay between narrative and mentalization. It signifies an engaging and valuable approach within the realm of behavioral research. We assessed mentalization skills through the Reflective Functioning Questionnaire after 4 months from the beginning of the school. After the assessment we provide the intervention for 5 months structured as follow: students wrote one essay per month (autobiographical narration) in which they talked about topics relate to personal life events. The narratives concerned the family, the relationship that students have with the world through a description of their own person, the perception of others, friendship, and the importance of social interactions. After reading to the class their essays, there was a discussion (2 h) in which students brainstormed on their own essays and others (Table 1).

The selection of themes appears to have been chosen to explore various dimensions of social relationships, self-perception, and the impacts of social interactions on individual life. In particular, significant life-long family interactions: family dynamics play a pivotal role in individual development and can influence psychological well-being over the course of a lifetime. Analyzing life-long family interactions can provide insights into how these dynamics change over time and impact individual growth; description of themselves: understanding how individuals describe themselves can unveil crucial aspects of their identity, self-esteem, and self-perception. This analysis may be valuable in exploring the consistency or evolution of self-perceptions over time; the others: expectations and criticality: the expectations of others and how individuals are criticized can influence self-esteem, behavior, and relationships. Exploring these aspects can contribute to understanding how social dynamics impact individual behavior and emotional well-being; friendships: reliable and unreliable friends: friendships play a significant role in people’s lives. Examining the differences between reliable and unreliable friendships could help identify key elements contributing to successful relationships and pinpoint factors that may negatively impact friendships; the value of social interactions: social interactions can profoundly impact mental health and overall well-being. Examining the value of social interactions may contribute to understanding how relationships influence various aspects of life, such as stress levels, satisfaction, and the provision of social support to navigate challenges. Finally, at the end of the school we assessed mentalization skills through the Reflective Functioning Questionnaire (RFQ_8).

## 3. Results

The data analysis, conducted using SPSS 26.0, revealed compelling insights into the impact of the intervention on participants’ mentalization skills. The choice of a stringent significance level (1%, α < 0.001) underscores the robustness of the statistical findings. The utilization of a two-way ANOVA with repeated measurements allowed for a comprehensive examination of changes in mentalization skills over the course of the intervention. Then, we performed a two-way ANOVA with repeated measurements (within = time factor at 2 levels) and dependent variable at 2 levels (scale = RFQ_C and RFQ_U).

The analysis showed the following results:

Interaction scale × time is significant [F (1.79) = 107.599, *p* < 0.001]. These data indicate that there is a significant interaction between the two subscales and time (Table 2).

We compared all subjects at T0 and T1 (within variable—time) to assess whether there were improvements in mentalization skills (RFQ_C and RFQ_U) after the intervention. The most pivotal finding was the significant interaction between the two subscales (RFQ_C and RFQ_U) and time (scale × time) with an F value of 107.599 and a significance level of *p* < 0.001. This interaction indicates that the intervention had a substantial effect on the relationship between certainty about mental states (RFQ_C) and uncertainty about mental states (RFQ_U) across the two time points (T0 and T1). This is a crucial result as it suggests that the intervention influenced the balance between hypermentalization and hypomentalization tendencies. Specifically, the observed improvement in mentalization skills post-intervention is noteworthy. At the outset (T0), participants exhibited a pattern where RFQ_U scores were low, indicating low uncertainty, while RFQ_C scores were high, indicating high certainty. This pattern saw a significant shift at T1, with both RFQ_U and RFQ_C tending to normalize. This normalization implies a more balanced and nuanced approach to mentalization, reflecting a reduction in the initial heterogeneity of participants’ mentalization skills. The graphical representation (see Figure 1) visually reinforces these findings, providing a clear depiction of the observed changes in RFQ_C and RFQ_U scores at T0 and T1. The graph illustrates the initial disparity at T0 and the subsequent convergence at T1, supporting the statistical significance of the interaction. These results hold substantial implications for the effectiveness of the intervention. The intervention not only led to an overall enhancement of mentalization skills but also contributed to a more harmonized and consistent pattern of mentalizing among participants. This suggests that the intervention had a lasting impact on the participants’ ability to understand and interpret mental states. In conclusion, the comprehensive analysis of the data supports the efficacy of the intervention in fostering positive changes in mentalization skills. These findings contribute valuable insights to the field, emphasizing the importance of targeted interventions in promoting balanced and adaptive mentalizing processes. Future research and interventions may benefit from considering these results to further refine and tailor strategies aimed at enhancing mentalization skills in diverse populations.

The use of a two-way repeated measures analysis is a robust methodology that allows examining the intervention’s effect over time and comparing changes in mentalization skills between the two periods, T0 and T1. The most significant result is the significant interaction between the two subscales (RFQ_C and RFQ_U) and time (scale × time) with an F value of 107,599 and a significance level of *p* < 0.001. This result confirms that the intervention had a significant impact on participants’ mentalization skills over time. Furthermore, the analysis revealed that at T0, participants had heterogeneous levels of mentalization skills: RFQ_U was low, while RFQ_C was high. This discrepancy was significantly reduced at T1, where both subscales showed substantial improvement, tending to normalize. This suggests that the intervention contributed to a balance in participants’ mentalization skills, leading to greater uniformity and consistency in their responses.

These results clearly indicate the effectiveness of the intervention in improving the mentalization skills of the participants in the study. This information can be valuable for future similar interventions and contribute to a deeper understanding of the dynamics of mentalization and its development over time.

## 4. Discussion

In this study, we have subjected the students to a teaching based on what Bruner theorizes as a narrative approach. The narratives that were exposed by the students referred to life stories, as a storytelling but of a more intimate and personal nature. We know how much storytelling can be therapeutic, even in many pathologies [20]. In addition, there are various types of narratives and narrators. For example, one is the restitution narrative that has a passive voice. The people who tell do not have a real exchange with the listener. Another type of narrative is one that speaks with a proactive voice and requires a listener who is ready to hear it and actively interact with the narrated story [21]. The latter is precisely the type of narration to which the sample students of the study have performed. Through activities based on the direct exchange of personal stories, students have had the opportunity to empathize with each other and make active reflections through the exploration of many different points of view. The outcomes of this study shed light on the positive effects of the narrative approach intervention on participants’ mentalization skills. The utilization of narrative techniques has long been recognized for its potential to facilitate self-reflection, empathy, and understanding of others’ perspectives. Our study reaffirms the efficacy of this approach, demonstrating significant enhancements in mentalization abilities among participants. Mentalization, as the theory of mind, encompasses a range of competencies that together enable people to solve problems in a world full of other minds. Examples include our ability to infer people’s emotions, understand others’ visual perspectives, trace behavior to actors’ underlying intentions and desires, and grasp the fact that others’ beliefs might misportray reality [12,13,14,15,16,17,18,19,20,21,22,23,24,25]. Adults also possess mentalization competencies that are less reliant on perceptual expertise. For example, most people are adept at taking others’ visual perspectives—that is, at imagining what the world looks like from a location not their own. This critical ability allows perceivers to infer what others can and cannot see, and thus do and do not know. Visual perspective taking is fully developed by the end of the preschool years [26,27] and can even be observed in some nonhuman primates [28]. Perspective-taking errors imply a failure to “select” another’s perspective when one should—for instance, when correct judgment requires perceivers to appreciate the difference between what they can see and what others can see [29]. The observed results not only validate the effectiveness of the narrative intervention but also underscore its transformative impact on participants’ mentalization skills. The significant interaction between the narrative approach and time (scale × time) emphasizes the dynamic nature of mentalization, which can be cultivated and refined through targeted therapeutic methods. Participants, who initially exhibited varying levels of reflective functioning (RFQ_U) and understanding (RFQ_C) at T0, experienced a notable convergence of these skills after the intervention. This convergence indicates the narrative approach’s capacity to bridge gaps in mentalization abilities, fostering a more balanced and nuanced understanding of self and others. The fundamental premise of the educational process should embrace an inherent openness, acknowledging the child or student not solely as a recipient upon whom we enforce specific behavioral norms, but also as an individual capable of nurturing and evolving their unique talents and capabilities [30]. The concept of identity within a narrative viewpoint involves constructing one’s autobiography by reflecting on their actions, emotions, and personal relationships in the framework of life events. This involves combining these interpretations to create a cohesive life story—a narrative comprised of multiple plots. The resemblance of identity formation to a story validates the application of narrative research methods [31]. Referring to this, we wanted to investigate how the perspective-taking implemented through the narrative approach could somehow improve the mentalization skills perceived by children over time. Indeed, as our results shown, we found out that there is a significant improvement in mentalization skills after the intervention. These findings offer a clear and articulate insight into how the power of stories can go beyond surface narration, significantly influencing how we learn, interact, and grow. A fundamental outcome of this study is the emphasis on the importance of integrating the narrative approach and mentalization processes in the educational context. The strategic use of stories and narratives not only enriches students’ learning process, but also fosters greater self-awareness and awareness of others. Moreover, promoting mentalization through narrative analysis significantly contributes to the development of crucial skills such as perspective-taking and emotional intelligence. Additionally, these findings contribute substantially to the evolving landscape of mentalization research. They underscore the adaptability of human mentalization capacities, demonstrating that targeted interventions can reshape these abilities positively. The narrative approach, by encouraging individuals to construct coherent and meaningful personal narratives, empowers them to navigate complex social situations with greater insight and empathy. This malleability in mentalization skills challenges traditional views that considered these abilities as relatively fixed traits, opening new avenues for therapeutic interventions and further research.

## 5. Conclusions

In the conclusion of this study on the narrative approach and mentalization, the extraordinary potential of these methodologies in the realms of education and human development becomes evident. The in-depth analysis of how narratives can serve as a vehicle for profound understanding of human experiences and how mentalization opens the doors to greater empathy and comprehension of others is enlightening. Finally, our study aims to encourage teaching based on strategies and mentalization processes in such a way as to increase the perception that students have of the latter in order to improve the skills of perspective-taking that would favor an improvement of the individual’s social skills. However, it is crucial to acknowledge the limitations inherent in our study. One notable limitation pertains to the absence of a follow-up component, which would have been instrumental in assessing the stability and enduring impact of the variables studied over the long term. The absence of long-term tracking poses a challenge in comprehensively evaluating the sustained effectiveness of the implemented strategies and mentalization processes. Another limitation is that the RFQ has been developed to assess severe impairments or imbalances in mentalizing as typically observed in patients with borderline personality disorder features. Hence, we advise researchers to use and validate the measure in samples of individuals that are likely to have severe problems with mentalizing, or in samples where at least enough variance in mentalizing capacities can be expected. Therefore, the measure might not be particularly suitable for use in normal community samples or student samples. Despite these limitations, the insights gained from our study provide a valuable foundation, offering valuable directions for future research and pedagogical interventions aimed at enhancing social cognition and interpersonal skills among students. In conclusion, this study serves as a foundational stepping stone to explore the depths of the human mind through the lenses of storytelling and mentalization. We hope that the discoveries and insights presented here can inspire further research and educational interventions that will shape the future of education and psychological well-being, highlighting the richness of our stories and the depth of our mutual understanding. In conclusion, this study underscores the importance of recognizing the limitations inherent in the absence of a control group. While the observed enhancement in mentalization skills is promising, it remains essential to acknowledge the possibility that the positive outcomes could be attributed to the influence of pleasant social interactions. Furthermore, addressing age-related variations and gender disparities is imperative for a comprehensive understanding of the intervention’s effectiveness across diverse demographics. Addressing these aspects not only enhances the validity and applicability of our findings but also provides valuable directions for future research and practical implementation in educational settings.

## Figures and Tables

**Figure 1 behavsci-13-00994-f001:**
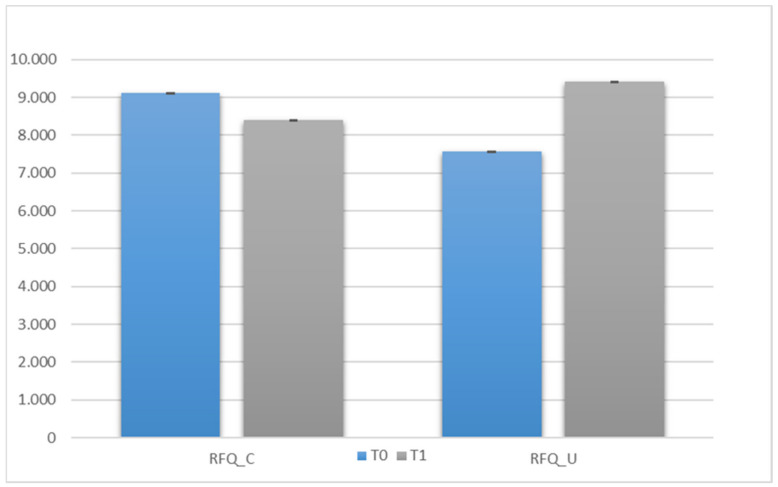
Comparison of the RFQ between T0 and T1.

**Table 1 behavsci-13-00994-t001:** A schematic view of the educational intervention.

Topic of the Essay	Duration	Discussion Modality
Significant life-long family interactions	3 h	Brainstorming (2 h)
Description of themselves	3 h	Brainstorming (2 h)
The others: expectations and criticality	3 h	Brainstorming (2 h)
Friendships: reliable and unreliable friend	3 h	Brainstorming (2 h)
The value of social interactions	3 h	Brainstorming (2 h)

**Table 2 behavsci-13-00994-t002:** Interaction between time × scale.

Time	Scale	Means	SD	F	P
0	RFQ_C	9.125	1.91		
	RFQ_U	7.575	0.63		
1	RFQ_C	8.413	0.49		
	RFQ_U	9.425	0.49	107,599	<0.000 *

* Statistical significance.

## Data Availability

The data presented in this study are available on request from the corresponding author.

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
