# Peer review of "Narrative Approach and Mentalization"

_behavsci, 2023, doi:10.3390/bs13120994_

Round 1
Reviewer 1 Report
Comments and Suggestions for Authors
The manuscript titled “Narrative approach and Mentalization” seeks to discuss combined approach to study mentalization and narration in a high school setting where 80 students from four different high schools were exposed to a particular teaching intervention. The manuscript, in its current form, is clear and detailed in some aspects. However, this review raised several issues that the authors should consider before this manuscript is ready to be published.
First, the current title of the manuscript could be revised to be more descriptive of the work that was actually done. Here are two suggestions
1. Harnessing the Power of Storytelling: Teaching Mentalization to High School Students through a Narrative-Based Intervention
2. Narrative-Based Teaching Interventions to Cultivate Mentalization in High School Students:
Second, the introduction is a block of text that spans three pages of the entire manuscript. Within this block of text, there are several topics and constructs that are in that are introduced. However, the manner in which they are connected is missing. Several topics such as mentalization, narrative development, mathematical thinking, historical references, are mentioned. This section of the paper would greatly benefit from an organizational structure. The authors might consider what are the topics that are going to be covered, how can they be tied together through subheadings, what would make this section easier for the readers to follow? In this section the overall goal/purpose of the study could be identified. Are there specific categories of literature that can be included here? Surprisingly, given the length of this section, there is a really small amount of references cited in these three pages. Thus, another suggestion would be for the authors to really engage with prior literature to expand the scope of their search somewhat to include more relevant and appropriate literature. This section may also include any guiding theory, theoretical or conceptual framework that is being used to guide the study.
Additionally, a more explicit connection needs to be made across all the topics and constructs included in the introduction so that the reader knows not only what topics are being covered in this section, but how they are connected.
In the Materials and Methods section, there are no research questions, research statement or hypotheses guiding the study identified. The participants are briefly described. However, the various contexts in which these participants exists would need further justification. Four high schools are mentioned but why are these four? How were these four schools selected? What connections might the researchers have had to the schools? How did they gain access? These are information one would expect to see in describing the participants and the context of the study.
Several constructs were also mentioned here, but were not clearly described. For example, they are talked about the inclusion criteria having QI between 95 to 105 through WAIS-IV. However, no explanation was given as to what this is, how or when this measure was assessed, and by whom.
Another protocol or tool mentioned was DSM-5 criteria, again no explanation of what this is was given. Several acronyms were introduced, but they were never explained, nor were they justified as to why they are appropriate for the study. Towards the end of the participants section, these measures were briefly described, but again, having some type of background information on how these were designed and or developed, how they were applied in the study, and how they were relevant for this study, would go a long way in helping the reader to understand not just what is being done, but why and how it will further our understanding of the goals of this study.
Three instruments are mentioned in the section, the SES, RFQ-8, and the WAIS-IV. Again, the authors should consider what justification would be necessary to help the reader to understand not only that these measures are appropriate, but how they were being assessed. What background information would someone need to have to confidently say that these measures are being applied appropriately, are appropriate for the study, and will actually return the kind of information necessary to meet the objectives/purpose of this work?
The procedures section is unclear. It is not clearly described how the procedure was conducted by whom or at what time? How were participants were incentivized or recruited to participate? Who administered the interventions? How often was the intervention administered? Did students require any type of priming activity to be able to participate in the intervention? Was the research procedure part of a class, course, or program?
The results section was also very short and underdeveloped. Some statistical tests are made are mentioned. However, the manner in which they were executed, the types of treatments that might have been applied to the data before the tests were conducted, who conducted the tests, how was quality of the data maintained is not clear, nor described. More details are needed about the actual findings of the study. Was the purpose of the study achieved? How were determinations of significance made? The authors may consider using descriptive statistics or other types of reporting strategies to clearly communicate the overall findings of the study. The section, as it is written, does not provide enough information to the reader or this reviewer as to what they found. This is also tying into the point about there being no research questions or hypotheses. Hence, what exactly these results are demonstrating is unclear.
The discussion section, like the results section, is very short. The focus of the study, also based on this discussion, seems incongruent with the method of execution, meaning, if this work is focused on mentalization and narrative development, then one could assume that there would be more explanations about the students’ development of narrative practices and demonstration of mentalization during or after the teaching intervention. Instead, the discussion reads as an extension of the introduction/background section, instead of an interpretation of the results. The purpose of the study is not mentioned nor how the results sit into the broader design of the study. Communicating what is the most important part of any manuscript i.e. how the work confirms or dispel previous work is not done here.
The conclusion section is clear and limitations of the study are made in an acceptable manner. However, there were several broad claims made in the introduction section about how groundbreaking this work is, how novel and insightful the findings will be and that there was a correlation between Bruner's narrative approach and mentalization. Additionally, similar claims made in the results section, such as the fundamental outcome of this study being to emphasize the importance of integrating narrative approach and mentalization processes was not accomplished.
Overall, this topic is interesting and work that seeks to understand students’ ability to narrate experiences is extremely important. However, the manuscript in its current form does not demonstrate this critical area of research.
Reviewer 2 Report
Comments and Suggestions for Authors
1. The section abstract is unnecessary to write the history of Mentalization and narrative. Hence, from line 11 to line 17, these sentences must be rewritten.
2. From line 24 to line 165, there is only one paragraph. The paragraph is too long length.
3. The section of the results must be rewritten. Each sentence is to form a paragraph.
4. The authors do not give the definitions of RFQ_C and RFQ_U.
5. The discussion content does not come from the section of the results.
Comments on the Quality of English LanguageModerate editing of English language required
Reviewer 3 Report
Comments and Suggestions for Authors
1. What is the main question addressed by the research?
How practical and theoretical activities based on re-elaborating information and events of personal life can improve the mentalization skills.
2. Do you consider the topic original or relevant in the field? Does it address a specific gap in the field?
Very specific. We have not enough biographical, qualitative research, especially in adult education.
3. What does it add to the subject area compared with other published material?
I think the topic is brand new, although the similar concepts are well developed, named in other way.
4. What specific improvements should the authors consider regarding the methodology? What further controls should be considered?
No comments.
5. Are the conclusions consistent with the evidence and arguments presented and do they address the main question posed?
Yes.
6. Are the references appropriate?
The text could be enriched with:
- The works of Gisela Labouvie-Vief and Manfred Spitzer (the theoretical approach). I am not sure if the works of M. Spitzer have been translated into English- the author could not be able to study in German or Spanish (I have found a Spanish translation only).
Also with articles:
Iwaszuk, M. (2019). Psycho-semiotic model of thinking. Combining Klein and Peirce theory of symbol for more comprehensive model of mind. Journal of Education Culture and Society, 10(1), 51–67. https://doi.org/10.15503/jecs20191.51.67
Pstross, M., Talmage, C. A., Peterson, C. B., & Knopf, R. C. (2017). In search of transformative moments: Blending community building pursuits into lifelong learning experiences. Journal of Education Culture and Society, 8(1), 62–78. https://doi.org/10.15503/jecs20171.62.78
Kondrla, P., Maturkanič, P., Taraj, M., & Kurilenko, V. (2022). Philosophy of Education in Postmetaphysical Thinking. Journal of Education Culture and Society, 13(2), 19–30. https://doi.org/10.15503/jecs2022.2.19.30
Jakubowska, L. . (2010). Identity as a narrative of autobiography. Journal of Education Culture and Society, 1(2), 51–66. https://doi.org/10.15503/jecs20102.51.66
7. Please include any additional comments on the tables and figures.
No comments.
Reviewer 4 Report
Comments and Suggestions for Authors
This is a brilliant project exploring narrative approach and mentalization in the context of educational practice. The authors have conducted a thorough literature review which builds on the work of Jerome Bruner and others. The methodological approach is comprehensive including both qualitative and quantitative aspects. The results section could have been enriched by including snippets of participants' experiencing of the narrative and mentalisation process. Otherwise the authors should be congratulated for this very important work which opens up new insights of pedagogical approaches which will enable students to experience education that enhances their 'social cognition' and 'interpersonal skills'.
Reviewer 5 Report
Comments and Suggestions for Authors
The paper addresses an original and important topic and has potential for a great contribution to scholars and practitioners, however certain revisions are needed so it fulfills its potential.
Although the authors made nice transitions between the concepts in the theoretical introduction, it is still difficult to follow the text. Therefore, the authors are advised to shorten some parts (e.g., about the narrative) or consider introducing some subtitles or at least formatting the text so it is easier to grasp all the ideas.
A short description of the instruments used is needed, particularly of the RFQ-8, including its reliability, examples of the items, range of scores. The Intervention has also to be described in more detail - what were concrete instructions and prompts, whether topic were predefined (if yes - with what idea) or students had a freedom to write whatever they wanted, how big the groups were, was there any dropout (and why), who and how facilitated the brainstorming and with what purpose, what was the group dynamic, where and when specifically these gatherings took place, etc. It is important that anyone who would like to try out this intervention knows exactly what steps to undertake and what to expect from participants. Similarly, the procedure of data collection and analyses could be described in more detail. Specifically, it is not clear what C and U refer to. Figure 2 does not convey any new information, so authors might consider removing it.
Although the focus of the paper is the effect of intervention, it would be useful to know more about the content and the form of narratives.
Discussion should include less sole theoretical points and more associating the results with theory. I assume some qualitative data (from the discussions, anecdotal notes, reflective journals) was also collected. That would help readers better understand the way participants reacted to the Intervention and what specifically in that intervention had positive effects on mentalization.
Authors need to acknowledge the limitation of not having a control group. We might assume that just having a pleasant social activity improved mentalization. They should also address the issues of age (is their conclusion applicable to all ages and if not - what night be the challenges) and gender (are there any differences). Some more specific guidelines for educators are also needed - what are all types of intervention that teachers could use, can they integrate these interventions in their classes when they have limited amount of time, etc.
Round 2
Reviewer 1 Report
Comments and Suggestions for Authors
Thank you for addressing my comments. I find the additions to be appropriate though the results section could benefit from a more detailed representation of the study's findings.
Reviewer 5 Report
Comments and Suggestions for Authors
Authors acknowledged the majority of the suggestions; however, there is still need for some improvements.
As in the meantime I found this specificity of the instrument the authors used, I recommend them to note that in the Limitations section:
"The RFQ has been developed to assess severe impairments or imbalances in mentalizing as typically observed in patients with borderline personality disorder features. Hence, we advise researchers to use and validate the measure in samples of individuals that are likely to have severe problems with mentalizing, or in samples where at least enough variance in mentalizing capacities can be expected. Therefore, the measure might not be particularly suitable for use in normal community samples or student samples."
As for this instrument, authors mentioned 8 items at one place and two scales with 6 items (12 in total) at the other, so they should make this uniform.
Although the authors included titles in the Figure 2, information about the intervention itself and justification of the selected topics and methods is still lacking. They should better answer to the Comment 2 directed to them before.
